# Markers of acute toxicity of DDT exposure in pancreatic beta-cells determined by a proteomic approach

Nela Pavlikova[1]*, Jan Sramek[1], Michael Jelinek[1], Petr Halada[2], Jan Kovar[1]

1 Department of Biochemistry, Cell and Molecular Biology & Center for Research of Diabetes, Metabolism, and Nutrition, Third Faculty of Medicine, Charles University, Prague, Czech Republic, 2 BioCeV–Institute of Microbiology, The Czech Academy of Sciences, Vestec, Czech Republic

☉ These authors contributed equally to this work.
* nela.pavlikova@lf3.cuni.cz

**Data Availability Statement:** All relevant data are within the manuscript and its Supporting Information file.

**Funding:** This work was supported by the research projects UNCE 204015 and PROGRES Q36 of

## Abstract

Many compounds have the potential to harm pancreatic beta-cells; organochlorine pollutants belong to those compounds. In this work, we aimed to find markers of acute toxicity of p,p'-DDT exposure among proteins expressed in NES2Y human pancreatic beta-cells employing 2-D electrophoresis. We exposed NES2Y cells to a high concentration (150 μM, $LC_{96}$ after 72 hours) of p,p'-DDT for 24 and 30 hours and determined proteins with changed expression using 2-D electrophoresis. We have found 22 proteins that changed their expression. They included proteins involved in ER stress (GRP78, and endoplasmin), mitochondrial proteins (GRP75, ECHM, IDH3A, NDUS1, and NDUS3), proteins involved in the maintenance of the cell morphology (EFHD2, TCPA, NDRG1, and ezrin), and some other proteins (HNRPF, HNRH1, K2C8, vimentin, PBDC1, EF2, PCNA, biliverdin reductase, G3BP1, FRIL, and HSP27). The proteins we have identified may serve as indicators of p,p'-DDT toxicity in beta-cells in future studies, including long-term exposure to environmentally relevant concentrations.

## Introduction

Many compounds have the potential to harm pancreatic beta-cells and disrupt glucose homeostasis in the human organism [1]. Such compounds include pharmaceuticals like pentamidine [2], or fluoxetine (SSRI antidepressant) [3] or saturated fatty acids palmitate [4], or stearate [5], and potentially also organochlorine pollutants, such as the now-banned pesticide DDT [6, 7]. Even decades after most countries banned its use, DDT and its metabolites persist in the environment [8, 9] and represent a threat to living organisms [10, 11]. Nowadays, DDT in human serum/plasma/blood commonly range between 1–500 nM [12, 13] with maxima occasionally overcoming 1 μM [14]. Epidemiologic studies [15–18] showed a correlation between DDT in the human organism and the incidence of diabetes mellitus. Nevertheless, they did not specify if DDT affected insulin production by pancreatic beta-cells or insulin signaling in target tissues [7, 19, 20].

Charles University in Prague, Czech Republic, and by the project BIOCEV CZ.1.05/1.1.00/02.0109 from the European Regional Development Fund. The funders had no role in study design, data collection, and analysis, decision to publish, or preparation of the manuscript.

In our previous study, we used 2-D electrophoresis coupled to mass spectrometry to find proteins possibly involved in mechanisms mediating a prolonged (1 month) effect of non-lethal concentrations of organochlorine pollutant *p,p'*-DDT in pancreatic beta-cells [6, 21]. In our present study, we aimed to find proteins that change expression in NES2Y human pancreatic beta-cells when exposed to a high concentration of *p,p'*-DDT) and could be detected by 2-D electrophoresis. Such proteins would represent markers of acute toxicity of DDT exposure in NES2Y human pancreatic beta-cells. They could be used to evaluate the effects of lower, environmentally more relevant concentrations of *p,p'*-DDT on pancreatic beta-cells. We also aimed to discuss the possible role of the changed expression of detected proteins in the damage caused to pancreatic beta-cells by exposure to a high concentration of *p,p'*-DDT. To achieve that, we exposed NES2Y human pancreatic beta cells to 150 μM concentration of *p,p'*-DDT for 24 and 30 hours and analyzed proteins with a changed expression using a proteomic approach (2-D electrophoresis coupled to MALDI-TOF mass spectrometry).

## Material and methods

### Material

We purchased *p,p'*-DDT (1,1,1-Trichloro-2,2-bis(4-chlorophenyl)ethane; product number 31041-100MG) from Sigma-Aldrich (www.sigmaaldrich.com), and propidium iodide from Abcam (www.abcam.cz: ab14085). For the western blot analysis, we used the following primary and secondary antibodies: anti-cleaved caspase-6 (#9761), anti-cleaved caspase-7 (#9491), anti-cleaved caspase-8 (#9496), anti-cleaved caspase-9 (#9505) anti-cleaved and total PARP (#9542), anti-GRP78 (#3177), and anti-CHOP (#2895) from Cell Signaling Technology (www.cellsignal.com). We purchased anti-actin (clone AC-40) primary antibody from Sigma-Aldrich (www.sigmaaldrich.com: A4700), and corresponding horseradish peroxidase-conjugated secondary antibodies from Proteintech (www.ptglab.com: SA00001-2, and SA00001-1).

### Cell culture

The NES2Y human pancreatic β-cell line was kindly provided by Dr. Roger F. James (Department of Infection, Immunity and Inflammation, University of Leicester) [22]. We routinely cultured NES2Y cells in a medium based on RPMI 1640, which contained penicillin (100 U/ml), streptomycin (100 μg/ml), sodium pyruvate (110 μg/ml), extra L-glutamine (300 μg/ml), HEPES (15 mM), and phenol red. We also supplemented the medium with 10% fetal bovine serum (FBS). We regularly test the cells for mycoplasma when we thaw the frozen cells. We passaged the cells twice per week. We routinely maintained cells in a humidified atmosphere of 5% CO2, in the air, at 37˚C 48. For the experiments, we used cells with a passage number between 15 and 20.

### Viability of cells

We seeded the cells in a 24-well plate in a concentration of 100 000 cells / 250 μl / well. After 24 hours, we exposed the cells to a scale of *p,p'*-DDT concentrations, i.e., 100 μM, 125 μM, 150 μM, 175 μM, 200 μM, and to DMSO (solvent control). The concentration of DMSO in the final media was 0.5%. After 24 hours, we harvested the cells by centrifugation (2000 rpm, 9 min, 4˚C). We resuspended cell pellets in a staining buffer containing propidium iodide (PI; dilution 1:100, ab14085, Abcam, Cambridge, UK) and incubated them for 10 minutes at room temperature in the dark. To detect the propidium iodide signal (emission = 585 nm) in dead cells, we used a signal detector FL2 of the FACS Calibur cytometer (https://www.bdbiosciences.com). We performed this experiment for three independent sets of samples.

## Western blot analysis

We seeded the cells (approximately 1 000 000 cells into a 50 mm Petri dish). After 24 hours of cultivation, we replaced the culture medium with a medium containing *p,p'*-DDT at a concentration of 150 μM. After 24 or 30 hours of incubation, we harvested the cells, lysed them, and quantified the protein concentration in samples employing the BCA method. For western blot analysis, we mixed 7.5 μl of samples containing 20 μg of proteins with 7.5 μl of sample loading buffer (0.125 mM Tris-HCl, pH 6.8, 10% glycerol, 4% SDS, 250 mM DTT, 0.004% bromphenol blue), heated for 7 min at 95˚C and then quickly cooled on ice. We separated proteins using a 10% polyacrylamide gel (with 4% polyacrylamide stacking gel) at 30 mA. We then blotted them onto a 0.2 μm nitrocellulose transfer membrane (Protran BA83, Schleicher-Schuell, Dassel, Germany) for 3 h at 0.25 A using a Mini-Protean 3 apparatus (www.bio-rad.com). We blocked the membrane for 20 min using 5% non-fat milk in TBS (100 mM Tris-HCl, 150 mM NaCl, pH 7.5) and then washed the membrane with 0.1% Tween-20/TBS three times. We incubated the washed membrane with respective primary antibodies in 0.1% Tween-20/TBS containing 1% non-fat milk overnight at 4˚C [23]. We diluted all primary antibodies 1:1000. After the incubation with the primary antibody, we washed the membrane and incubated it with the corresponding horseradish peroxidase-conjugated secondary antibody for 2 h. After washing, we detected a secondary antibody signal by enhanced chemiluminescence employing a Carestream Gel Logic 4000 PRO Imaging System equipped with Carestream Molecular Imaging Software (www.carestream.com). We employed Image Master™ 2D Platinum 6.0 software (www.gehealthcare.com) to obtain data for densitometric analyses.

## 2-D electrophoresis and subsequent gel analysis

We run 2-D electrophoresis for at least three independent sets of samples.

**Exposure to p,p'-DDT.**   For the experiment, we seeded 5 000 000 NES2Y cells into a medium-sized flask. After 24 hours of cultivation, we replaced the medium with a medium containing 150 μM *p,p'*-DDT, or DMSO. After 24 or 30 hours, we harvested the cells. The concentration of DMSO (solvent control) in the medium was 0.5%.

**Preparation of samples for 2-D electrophoresis.**   We trypsinized the cells, washed them 3x times with ice-cold PBS, and resuspended them in Protein Extraction Buffer-V ((GE Healthcare, www.gehealthcare.com)) (urea, thiourea, CHAPS) containing 2% of Protease Inhibitor Mix ((GE Healthcare, www.gehealthcare.com)). We purified all samples using a 2-D Clean-Up Kit (GE Healthcare, www.gehealthcare.com) following the manufacturer´s instructions. Consequently, we dissolved the proteins in Protein Extraction Buffer-V suitable for isoelectric focusing. We used the 2-D Quant Kit ((GE Healthcare, www.gehealthcare.com) weden) to determine protein concentrations.

**2-D electrophoresis: Isoelectric focusing.**   We used an IPGphor focusing unit (GE Healthcare, Uppsala, Sweden) for the isoelectric focusing of samples loaded on 7 cm pH 4–7 Immobiline DryStrips (GE Healthcare, Uppsala, Sweden). We rehydrated each strip for 48 hours, as described previously [24]. After rehydration, pH 4–7 strips were focused with a limited current of 50 μA/strip at 20˚C using the following conditions: gradient 0→150 V for 2 h, 150 V for 1 h, gradient 150→300 V for 1 h, 300 V for 2 h, gradient 300→1200 V for 3 h, 1200 V for 1 h, gradient 1200→3500 V for 5 h, and 3500 V for 5.5 h.

**2-D electrophoresis: SDS-PAGE.**   Following isoelectric focusing, we equilibrated the strips for 20 min in an equilibration buffer [25] containing 2% DTT. Then, we changed the buffer for a new one containing 2.5% iodoacetamide instead of 2% DTT and equilibrated strips for another 20 min. After equilibration, we placed the strips on top of the gels, sealed them using 0.5% agarose containing bromphenol blue, and run SDS-PAGE. We employed a Mini-PROTEAN Tetra cell (Bio-Rad, www.bio-rad.com) device for the second dimension and used

10% polyacrylamide gels with 4% stacking gels for separation. We run gels at a constant voltage of 50 V until the blue line reached the bottom of the gels (approximately 3 h). After running the second dimension, we washed each gel 3 x 5 min in distilled water and stained in 50 ml of colloidal Coomassie brilliant blue (CBB) solution [26] overnight.

**Gel image and analysis.** After staining, we scanned gels using a calibrated UMAX Power-Look 1120 scanner running LabScan software (GE Healthcare, www.gehealthcare.com). We used Image Master™ 2D Platinum 6.0 software (GE Healthcare, www.gehealthcare.com) to analyze the gels. We analyzed differences between corresponding spots in each set of gels (NES2Y exposed to DMSO and *p,p'*-DDT 150 μM). We selected spots with an approximately twofold (or bigger) difference in expression between the cell lysate exposed to DMSO and the cell lysate exposed to 150 μM DDT as spots with a different expression. We determined the statistical significance of changes in protein expression using the student´s t-test. Spots with significantly different intensities were cut and sent for MS analysis.

**Enzymatic digestion, MALDI-TOF mass spectrometry, and protein identification.** After destaining CBB-protein spots, we incubated them overnight at 37 ˚C in a buffer containing 25 mM 4-ethyl morpholine acetate, 5% acetonitrile, and trypsin (100 ng; Promega). We mixed the resulting peptides with a MALDI matrix consisting of an aqueous 50% acetonitrile/ 0.1% TFA solution of α-cyano-4-hydroxycinnamic acid (5 mg/ml; Sigma-Aldrich, www. sigmaaldrich.com). We measured mass spectra using an Ultraflex III MALDI-TOF (Bruker Daltonics, Bremen, Germany) in a mass range of 700–4000 Da calibrated externally using a mixture of PepMix II standard (Bruker Daltonics). We searched both MS and MS/MS data against the SwissProt 2017_03 database subset of human proteins using the in-house MAS-COT software with the following settings: peptide tolerance of 30 ppm, missed cleavage site set to one, variable carbamidomethylation of cysteine, and oxidation of methionine. We considered proteins with MOWSE scores over the threshold of 54 (calculated for the settings used) as identified. We confirmed the identity of each protein candidate using MS/MS analysis.

## Confocal microscopy

We seeded the NES2Y cells at a density of 60 000 cells / 0.5 ml of the medium onto coverslips. After 24 hours, we replaced the medium with a fresh one containing 150 μM *p,p'*-DDT. We used cells cultivated in a medium without pollutants as control cells.

After 24 hours, we discarded the medium and washed the cells three times with PBS (5–10 min). Then, we fixed cells using 4% paraformaldehyde for 20 min, washed them by PBS, permeabilized them with 0.3% Triton X-100 for 10 min, and washed them again. After that, we blocked cells with 1% BSA for 60 minutes and stained them with the primary anti-tubulin antibody (ab6046 from Abcam, Cambridge, UK) diluted 1:300 in 1% BSA, at 4˚C overnight. After washing the cells with PBS, we incubated the cells with a secondary goat anti-rabbit antibody (ab150077, from Abcam) diluted 1:300, in the dark at room temperature for 2 hours. To complete tubulin staining, we washed the cells again with PBS. Then, we stained actin in the cells by rhodamine-phalloidin (R415, from Invitrogen) diluted according to the manufacturer's instruction for 20 min. The unbound staining solution was removed by PBS washing. Finally, we transferred the cells onto a droplet of Vectashield® Vibrance™ Antifade Mounting Medium with DAPI (Vector Laboratories, Burlingame, CA, USA) and sealed them. Samples were analyzed using a Leica TCS SP5 confocal microscope (Bannockburn, IL, USA).

## Statistical analysis

We analyzed the statistical significance of the results of 2-D electrophoresis and western blot using a student´s t-test. We analyzed the statistical significance of the results of flow cytometry

using a one-way ANOVA Dunnett´s test (SigmaPlot 14 Software). We considered the differences with p < 0.05 as statistically significant.

## Results

### Effect of various concentrations of p,p'-DDT on the viability of NES2Y cells

For our study, we wanted to establish a concentration of *p,p'*-DDT that would kill approximately 20% of cells after 24 hours of exposure, so the remaining 80% of cells would provide us with enough proteins for 2-D electrophoresis. The tested concentrations were chosen based on our previous results [6]. After 24 hours of exposure, the two lower concentrations of *p,p'*-DDT (100 μM and 125 μM) caused only a non-significant decrease in cell viability when compared with the control (Fig 1). The three higher concentrations (150 μM, 175 μM, and 200 μM) caused a significant reduction of viability of cells (Fig 1), (p = 0.11 for 150 μM, p < 0.001 for 175 μM, p < 0.001 for 200 μM). For the planned experiments, we chose a concentration of 150 μM that reduced cell viability to 82% of the control after 24 hours of exposure. Further testing established a 150 μM concentration of *p,p'*-DDT as $LC_{96}$ (lethal concentration) for NES2Y after 72 hours (see S1 Fig in S1 File).

### Effect of p,p'-DDT on apoptotic and ER stress markers

In order to confirm that p,p'-DDT induced apoptotic cell death in NES2Y cells, we tested the presence of the main markers of apoptosis, i.e., activated caspase-8, -9, -6, -7 and cleaved PARP (poly (ADP-ribose) polymerase)—a substrate of caspase-7, (Fig 2, S4–S10 Figs in S1 File). We did not detect caspase-3 since we know that it is not functional in NES2Y cells [23]. The level of cleaved caspase-8 increased to 235% of the control (p = 0.039) after 24-hour exposure, and to 391% of the control (p = 0.015) after 30-hour exposure (Fig 3). The level of cleaved caspase-9 also increased after both 24-hour exposure (581% of the control, p = 0.018) and 30-hour exposure (423% of the control, p = 0.004), (Fig 3). The level of cleaved caspase-6 increased to 555% of the control (p = 0.042) after 24 hours of exposure, and to 635% of the

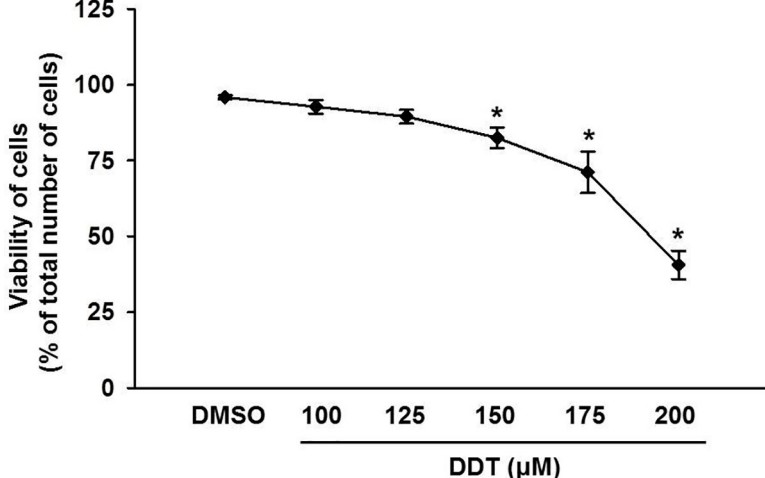

**Fig 1. Viability of NES2Y cells exposed to various concentrations of *p,p'*-DDT for 24 hours.** We determined the number of viable cells using flow cytometry after staining with propidium iodide. We used cells exposed to DMSO (final concentration of 0.5%) as solvent control. The graph represents the mean of results from 3 independent experiments. * means a statistically significant difference (p < 0.05) when compared to the solvent control (one-way ANOVA, Dunnett's test).

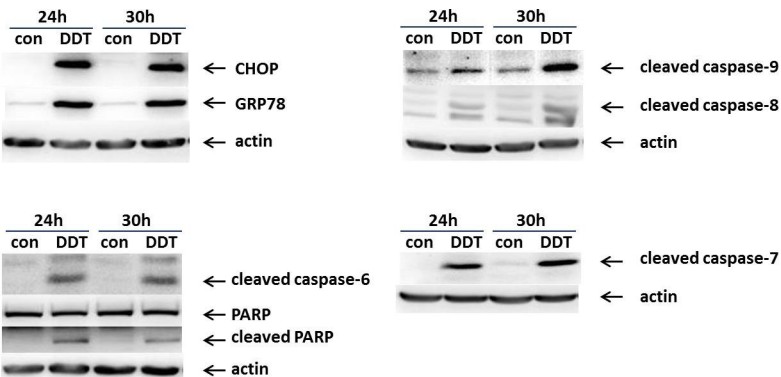

**Fig 2. Representative western blots of selected markers of ER stress (CHOP, GRP78) and apoptosis (cleaved caspase-6, -7, -8, -9, and PARP).** The picture shows the expression of chosen proteins in NES2Y cells exposed to DMSO as a solvent control (con) and 150 μM *p,p'*-DDT for 24 and 30 hours. We used actin as a loading control. CHOP means C/EBP homologous protein, GRP78 means 78kDa glucose-regulated protein, and PARP means poly (ADP-ribose) polymerase.

control ($p = 3.77 \times 10^{-4}$) after 30-hour exposure (Fig 3). The level of cleaved caspase-7 represented the highest increase; it reached 1434% of the control ($p = 0.003$) after 24-hour exposure, and after 30-hour exposure, it reached 1820% ($p = 0.008$) of the control. The level of cleaved PARP reached 240% of the control after 24-hour exposure ($p = 0.022$), and after 30-hour exposure, it reached 226% of the control ($p = 0.003$), (Fig 3).

To confirm that p,p'-DDT induces ER stress in NES2Y cells, we have tested the effect of p, p'-DDT on the expression of proteins BiP and CHOP—the main markers of ER stress. The level of CHOP increased to 1599% of the control ($p = 0.021$) after 24-hour exposure, and to

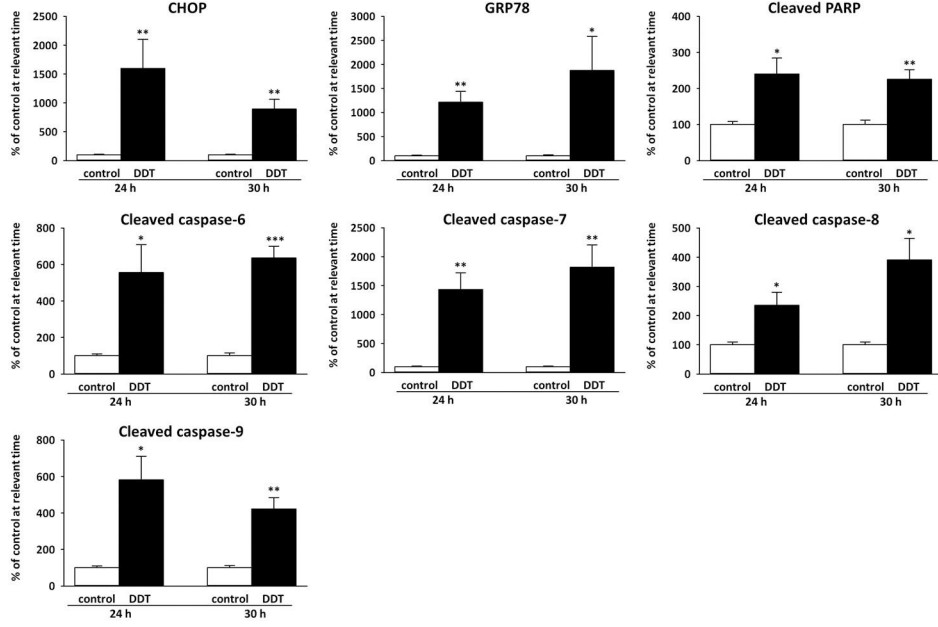

**Fig 3. Densitometry of western blots of chosen markers of ER stress and apoptosis.** Columns represent mean values ± SEM of protein levels from 3 independent sets of experiments. CHOP means C/EBP homologous *protein*, *GRP78 means 78kDa glucose-regulated protein*, and PARP means poly (ADP-ribose) polymerase. *, **, *** means a statistically significant difference with p<0.05, p<0.01, and p<0.001, respectively, when compared to the control at the relevant time (analyzed by the student's t-test).

893% of the control (p = 0.002) after 30-hour exposure. The expression of GRP78 was upregulated to 1213% of the control (p = 0.002) after 24-hour exposure, and to 1875% of the control (p = 0.034) after 30-hour exposure (Figs 2 and 3, S2 and S3 Figs in S1 File).

## Proteins with changed expression after 24 hours of exposure to p,p'-DDT

The 24-hour exposure to 150 μM p,p'-DDT changed the expression of 10 spots; 4 spots were upregulated, and 6 spots were downregulated when compared with control (Fig 4 and S11 Fig in S1 File, Table 1). 78 kDa glucose-regulated protein (GRP78, also known as BiP) was represented by three upregulated spots (GRP78*, GRP78**, GRP78***); spot GRP78* was upregulated to 1610% of the control (p = 0.045), spot GRP78** to 240% of the control (p = 0.003), and spot GRP78*** to 285% of the control (p = 0.044). Another glucose-regulated protein, 75 kDa

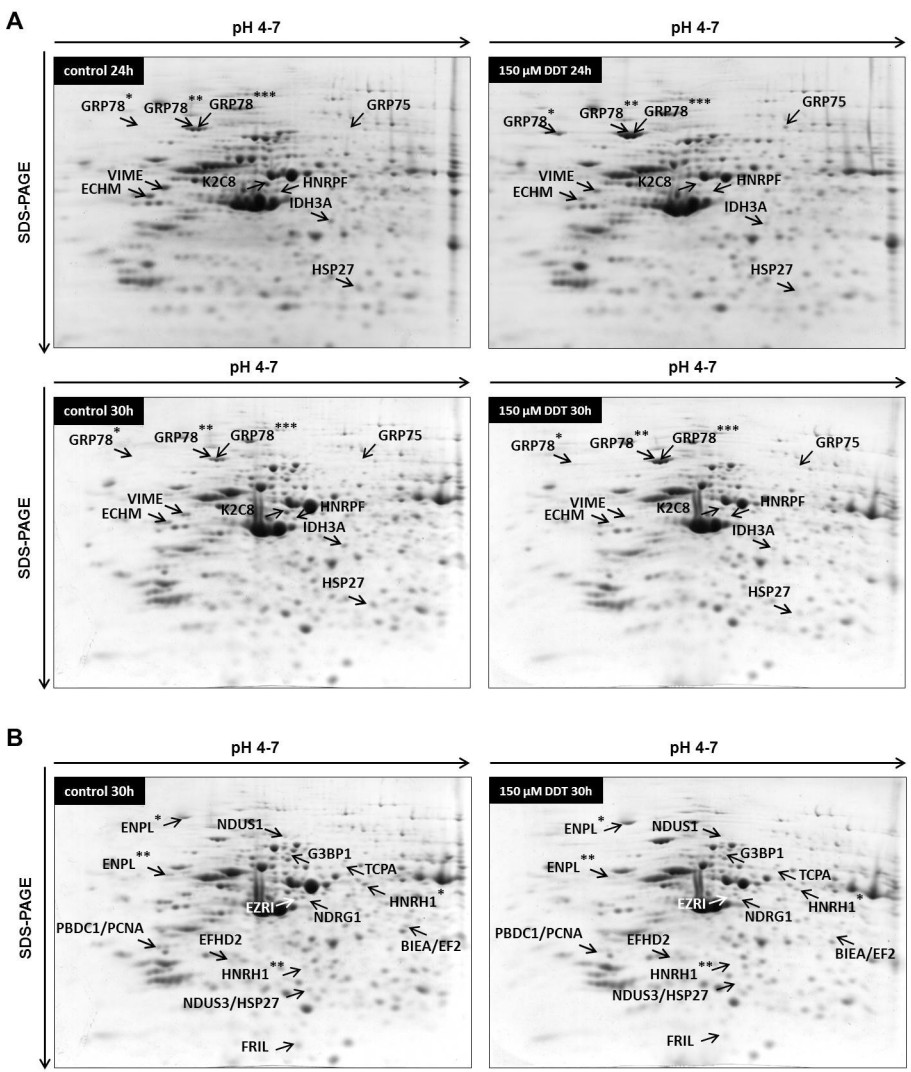

**Fig 4. Representative 2-DE gels (pI range 4–7) of NES2Y cells exposed to 150 μM p,p'-DDT in DMSO for 24 hours and 30 hours.** Part **A** shows spots/proteins that changed their expression after both 24-hour and 30-hour exposure. *, **, *** mark different spots identified as the same protein. Part **B** shows spots/proteins that changed their expression only after 30-hour exposure. *, ** mark different spots identified as the same protein. BIEA/EF2, NDUS3/HSP27, and PBDC1/PCNA represent spots that contained two different proteins. For the full names of detected proteins, see Tables 1 and 2.

**Table 1. Differentially expressed proteins after both 24-hour and 30-hour exposure to 150 μM *p,p'*-DDT identified in 2-DE experiments using three independent sets of samples.**

| Fold change—24 h | Fold change—30 h | Protein name | DTB No. | No. peptides | Coverage [%] | MS/MS confirmation | MW protein | pI |
|---|---|---|---|---|---|---|---|---|
| 0.44 | 0.39 | **ECHM** ↓ (enoyl-CoA hydratase, mitochondrial) | ECHM_HUMAN | 5 | 25 | ASGANFEYIIAEKR AQFAQPEILIGTIPGAGGTQR | 31 | 8.3 |
| 10.83 | 8.57 | **GRP75** ↑ (75 kDa glucose-regulated protein) | GRP75_HUMAN | 10 | 18 | VQQTVQDLFGR NAVITVPAYFNDSQR STNGDTFLGGEDFDQALLR | 74 | 5.9 |
| 16.10 | - - - | **GRP78**\* ↑ (78 kDa glucose-regulated protein) | GRP78_HUMAN | 13 | 22 | ITPSYVAFTPEGER VTHAVVTVPAYFNDAQR IEIESFYEGEDFSETLTR | 72 | 5.1 |
| 2.40 | 2.42 | **GRP78**\*\* ↑ (78 kDa glucose-regulated protein) | GRP78_HUMAN | 15 | 27 | ITPSYVAFTPEGER IEIESFYEGEDFSETLTR | 72 | 5.1 |
| 2.85 | 1.97 | **GRP78**\*\*\* ↑ (78 kDa glucose-regulated protein) | GRP78_HUMAN | 14 | 24 | ITPSYVAFTPEGER VTHAVVTVPAYFNDAQR IEIESFYEGEDFSETLTR | 72 | 5.1 |
| 0.48 | 0.50 | **HNRPF** ↓ (heterogeneous nuclear ribonucleoprotein F) | HNRPF_HUMAN | 6 | 17 | HSGPNSADSANDGFVR ATENDIYNFFSPLNPVR | 46 | 5.4 |
| 0.54 | 0.42 | **HSP27** ↓ (heat shock protein 27) | HSPB1_HUMAN | 9 | 43 | RVPFSLLR LFDQAFGLPR LATQSNEITIPVTFESR | 23 | 6.0 |
| 0.41 | 0.43 | **IDH3A** ↓ (isocitrate dehydrogenase [NAD] 3 subunit alpha, mitochondrial) | IDH3A_HUMAN | 5 | 17 | IAEFAFEYAR TIPIDGNFFTYTR | 40 | 6.5 |
| 0.41 | 0.26 | **K2C8** ↓ (keratin, type II cytoskeletal 8) | K2C8_HUMAN | 7 | 15 | LEGLTDEINFLR ASLEAAIADAEQRGELAIK | 54 | 5.5 |
| 0.51 | 0.36 | **VIME** ↓ (vimentin) | VIME_HUMAN | 16 | 43 | ISLPLPNFSSLNLR EMEENFAVEAANYQDTIGR QVQSLTCEVDALKGTNESLER | 54 | 5.1 |

The table covers the proteins that changed expression after both 24 hours and 30 hours of exposure. The table includes fold change after 24 hours and 30 hours, protein name, SwissProt database number, the number of peptides matched to the identified protein, sequence coverage, peptide sequences confirmed by MS/MS, theoretical molecular weight, and pI of the protein.

\*, \*\*, \*\*\* mark different spots identified as the same protein.

glucose-regulated protein (GRP75, also known as mortalin), was found upregulated too (spot GRP75, expression increased to 1083%, p = 3.38 x $10^{-4}$).

The 24-hour exposure to 150 μM p,p'-DDT downregulated the expression of enoyl-CoA hydratase mitochondrial to 44% of the control (spot ECHM, p = 0.006), the expression of vimentin to 51% of the control (spot VIME, p = 4.82 x $10^{-5}$), the expression of heat shock protein 27 to 54% of the control (spot HSP27, p = 0.026), the expression of isocitrate dehydrogenase [NAD] subunit alpha mitochondrial to 41% of the control (spot IDH3A, p = 3.84 x $10^{-5}$), the expression of keratin, type II cytoskeletal 8 to 41% of the control (spot K2C8, p = 3.09 x $10^{-8}$), and the expression of heterogeneous nuclear ribonucleoprotein F to 48% of the control (spot HNRPF, p = 0.021) (Fig 4, Table 1).

## Proteins with changed expression after 30 hours of exposure to p,p'-DDT

Almost all proteins that changed their expression after 24-hour exposure also changed expression after 30-hour exposure (Fig 4 and S12 Fig in S1 File, Table 1). The only exception was spot GRP78\* (78 kDa glucose-regulated protein). The spot was smeared and blended with surrounding spots; therefore, it could not be analyzed. The 30-hour exposure to 150 μM *p,p'*-DDT upregulated the expression of 78 kDa glucose-regulated protein to 242% of the control (spot GRP78\*\*, p = 7.59 x $10^{-8}$, and to 197% of the control (spot GRP78\*\*\*, p = 1.88 x $10^{-5}$),

and the expression of 75 kDa glucose-regulated protein to 857% of the control (spot GRP75, p = 7.75 x $10^{-4}$).

The 30-hour exposure to 150 μM *p,p'*-DDT downregulated the expression of more proteins than were upregulated. The expression of enoyl-CoA hydratase mitochondrial (spot ECHM, p = 3.91 x $10^{-5}$) decreased to 39% of the control, vimentin (spot VIME, p = 2.97 x $10^{-4}$) to 36% of the control, heat shock protein 27 (spot HSP27, p = 8.34 x $10^{-5}$) to 42% of the control, isocitrate dehydrogenase [NAD] 3 subunit alpha mitochondrial (spot IDH3A, p = 1.86 x $10^{-4}$) to 43% of the control, keratin type II cytoskeletal 8 (spot K2C8, p = 8.94 x $10^{-6}$) to 26% of the control, and heterogeneous nuclear ribonucleoprotein F (spot HNRPF, p = 1.11 x $10^{-4}$) to 50% of the control. The changes of expression remained quite similar after both exposure times in most of the proteins (see Table 1).

Besides 10 proteins with changed expression detected after both 24-hour exposure and 30-hour exposure, there were also 14 more spots with changed expression detected only after 30-hour exposure (Fig 4, Table 2). Those 14 spots represented 14 proteins; nevertheless, some spots contained two proteins, and some proteins occurred as two different spots. We found 3 proteins upregulated: N-myc downregulated gene 1 protein (spot NDRG1, p = 0.019) to 312% of the control, EF-hand domain-containing protein D2 (spot EFHD2, p = 3.01 x $10^{-3}$) to 274% of the control, and one of the two spots representing endoplasmin (spot ENPL*, p = 1.31 x $10^{-4}$) to 199% of the control (Fig 4, Table 2).

Downregulated proteins included Ras GTPase-activating protein-binding 1 (spot G3BP1, p = 1.57 x $10^{-4}$) with the expression downregulated to 53% of the control, NADH-ubiquinone oxidoreductase 75 kDa subunit mitochondrial (spot NDUS1, p = 1.71 x $10^{-8}$) with the expression downregulated to 25% of the control, and T-complex protein 1 subunit alpha (spot TCPA, p = 7.28 x $10^{-6}$) with the expression downregulated to 40% of the control. The expression of ferritin light chain (spot FRIL, p = 3.23 x $10^{-5}$) was downregulated to 49% of the control, and the expression of ezrin (spot EZRI, p = 1.11 x $10^{-4}$) to 50% of the control. The position of spot EZRI on the 2-D gel did not correlate with its predicted molecular; spot EZRI represents a fragment of ezrin. We identified two other spots as heterogeneous nuclear ribonucleoprotein H (HNRH1): expression of HNRH1** (a fragment of a protein) decreased to 33% of the control (p = 2.50 x $10^{-4}$), and expression of HNRH1* (a whole form of protein) to 53% of the control (p = 1.61 x $10^{-5}$). Expression of endoplasmin fragment (spot ENPL**, p = 3.69 x $10^{-5}$) decreased to 51% of the control (Fig 4, Table 2).

Several spots with downregulated expressions contained two proteins each. The spot with expression downregulated to 53% of the control (spot NDUS3/HSP27, p = 7.28 x $10^{-4}$) contained both NADH dehydrogenase [ubiquinone] iron-sulfur protein 3 mitochondrial and heat shock protein 27. The spot with the expression downregulated to 56% of the control (spot PBDC1/PCNA, p = 1.37 x $10^{-3}$) contained protein PBDC1 and proliferating cell nuclear antigen (PCNA). Furthermore, the spot with the expression downregulated to 43% of the control (spot BIEA/EF2, p = 3.11 x $10^{-4}$) contained both biliverdin reductase (BIEA) and elongation factor 2 (EF2) (Fig 4, Table 2). The position of the BIEA/EF2 spot did not correlate with the predicted size for EF2, which means that the spot contained an EF2 fragment.

## Morphology of the cells exposed to p,p'-DDT

To visualize changes in the morphology of cells exposed to p,p'-DDT (150 μM), we employed immunofluorescence of the cytoskeletal proteins actin and tubulin. We found no cells undergoing mitosis or cytokinesis among cells exposed to *p,p'*-DDT, but such cells occurred among cells exposed to solvent control (DMSO) (Fig 5). Many cells exposed to *p,p'*-DDT had a more elongated shape than the control cells. Some cells exposed to p,p'-DDT were even divided into two parts connected by a long thin "neck" and with a nucleus located in one of those parts.

**Table 2. Differentially expressed proteins identified in 2-DE experiments.**

| Fold change—30 h | Protein name | DTB No. | No. peptides | Coverage [%] | MS/MS confirmation | MW protein | pI |
|---|---|---|---|---|---|---|---|
| 0.43 | **BIEA** ↓ (biliverdin reductase A), | BIEA_HUMAN | 4 | 17 | FGFPAFSGISR GSLLFTAGPLEEER | 33 | 6.6 |
| | and **EF2** ↓ (elongation factor 2) | EF2_HUMAN | 3 | 4 | VNFTVDQIR ALLELQLEPEELYQTFQR | 95 | 6.4 |
| 2.74 | **EFHD2** ↑ (EF-hand domain-containing protein D2) | EFHD2_HUMAN | 6 | 27 | FEEEIKAEQEER VFNPYTEFKEFSR | 27 | 5.2 |
| 1.99 | **ENPL*** ↑ (endoplasmin) | ENPL_HUMAN | 13 | 18 | SILFVPTSAPR FQSSHHPTDITSLDQYVER | 92 | 4.8 |
| 0.51 | **ENPL**** ↓ (endoplasmin) | ENPL_HUMAN | 8 | 10 | FAFQAEVNR SILFVPTSAPR | 92 | 4.8 |
| 0.50 | **EZRI** ↓ (ezrin) | EZRI_HUMAN | 8 | 11 | QLLTLSSELSQAR RKEDEVEEWQHR | 69 | 5.9 |
| 0.49 | **FRIL** ↓ (ferritin light chain) | FRIL_HUMAN | 6 | 40 | DDVALEGVSHFFR LNQALLDLHALGSAR LGGPEAGLGEYLFER KLNQALLDLHALGSAR | 20 | 5.5 |
| 0.53 | **G3BP1** ↓ (Ras GTPase-activating protein-binding protein 1) | G3BP1_HUMAN | 6 | 17 | FYVHNDIFR | 52 | 5.4 |
| 0.53 | **HNRH1*** ↓ (heterogeneous nuclear ribonucleoprotein H) | HNRH1_HUMAN | 7 | 24 | GLPWSCSADEVQR HTGPNSPDTANDGFVR ATENDIYNFFSPLNPVR EGRPSGEAFVELESEDEVK | 49 | 5.9 |
| 0.33 | **HNRH1**** ↓ (heterogeneous nuclear ribonucleoprotein H) | HNRH1_HUMAN | 4 | 13 | ATENDIYNFFSPLNPVR | 49 | 5.9 |
| 3.12 | **NDRG1** ↑ (N-myc downstream regulated 1) | NDRG1_HUMAN | 3 | 14 | SIIGMoxGTGAGAYILTR GNRPVILTYHDIGMoxNHK | 43 | 5.5 |
| 0.53 | **NDUS3** ↓ (NADH dehydrogenase [ubiquinone] iron-sulfur protein 3, mitochondrial), | NDUS3_HUMAN | 9 | 31 | VVAEPVELAQEFR | 30 | 7.0 |
| | and **HSP27** ↓ (heat shock protein 27) | HSPB1_HUMAN | 6 | 31 | LFDQAFGLPR LATQSNEITIPVTFESR | 23 | 6.0 |
| 0.25 | **NDUS1** ↓ (NADH-ubiquinone oxidoreductase 75 kDa subunit, mitochondrial) | NDUS1_HUMAN | 10 | 15 | FEAPLFNAR | 79 | 5.9 |
| 0.56 | **PBDC1** ↓ (polysaccharide biosynthesis domain containing 1), | PBDC1_HUMAN | 5 | 25 | IQFFAIEIAR FNGIVEDFNYGTLLR | 26 | 4.7 |
| | and **PCNA** ↓ (proliferating cell nuclear antigen) | PCNA_HUMAN | 6 | 26 | SEGFDTYR CAGNEDIITLR | 29 | 4.6 |
| 0.40 | **TCPA** ↓ (T-complex protein 1 subunit alpha) | TCPA_HUMAN | 6 | 12 | EQLAIAEFAR | 60 | 5.8 |

The table covers proteins that changed expression ONLY after 30 hours of exposure. Some spots contained two different proteins. The table includes fold change after 30 hours, protein name, SwissProt database number, the number of peptides matched to the identified protein, sequence coverage, peptide sequences confirmed by MS/MS, theoretical molecular weight, and pI of the protein.

*, **, *** mark different spots identified as the same protein.

## Discussion

In this study, we used 2-D electrophoresis coupled to mass spectrometry to find markers of acute toxicity of DDT exposure in NES2Y human pancreatic beta-cells. Together, 2-D electrophoresis revealed 22 proteins with altered expression. We have sorted these proteins into groups based on their function and tried to evaluate the meaning and importance of changes in the expression of individual proteins.

### Proteins involved in the stress of endoplasmic reticulum

Strong upregulation of **78 kDa glucose-regulated protein** (**GRP78**, also known as BiP) indicated the presence of the stress of endoplasmic reticulum (ER stress) in cells exposed to *p,p'*-DDT [27, 28]. Another protein, **CHOP**, mediates the ER stress-induced apoptosis [29], and its

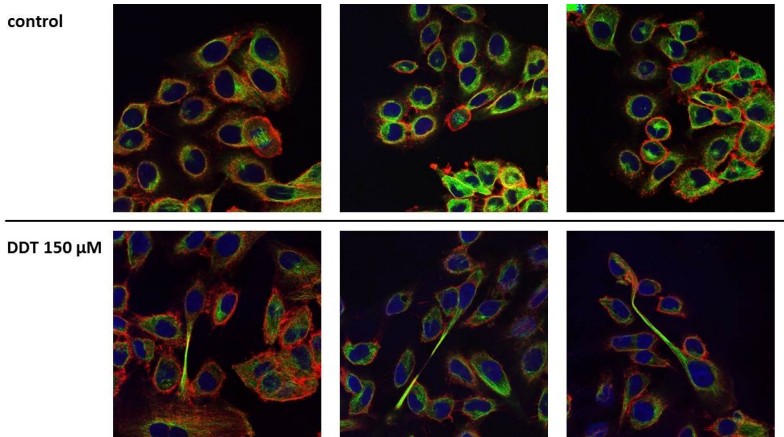

**Fig 5. Effect of 150 μM p,p'-DDT on the cell shape of NES2Y pancreatic beta-cells.** Cells incubated without *p,p'*-DDT represented control cells. After 24 hours of incubation, we stained actin (red), tubulin (green), and nuclei (blue) of the cells. The figure shows representative pictures.

upregulation in cells exposed to *p,p'*-DDT supports the idea that ER stress played a role in cell death induced by *p,p'*-DDT in pancreatic beta-cells.

The exposure to *p,p'*-DDT also increased the expression of **endoplasmin** (or heat shock protein 90 kDa beta member 1) in pancreatic beta-cells (see Table 2). Endoplasmin plays a vital role in cell survival under ER stress [30–32]. Therefore, its upregulation supports the idea that exposure to *p,p'*-DDT induces ER stress in beta-cells. Another spot identified as endoplasmin had a smaller size than predicted, and its expression was downregulated. Endoplasmin is a dimer [31], and theoretically, the downregulated spot could represent the endoplasmin monomer.

## Mitochondrial proteins

We found four mitochondrial proteins downregulated in pancreatic beta-cells after exposure to p,p'-DDT: ECHM, IDH3A, NDUS1, and NDUS3. **Enoyl-CoA hydratase** (**ECHM**) participates in β-oxidation, **isocitrate dehydrogenase [NAD] 3 subunit alpha** (**IDH3A**) in the citric acid cycle, **NADH-ubiquinone oxidoreductase 75 kDa subunit** (**NDUS1**), and **NADH dehydrogenase [ubiquinone] iron-sulfur protein 3** (**NDUS3**) belong to the complex I of the respiratory chain. The 24-hour exposure to *p,p'*-DDT reduced the expression of ECHM and IDH3; the 30-hour exposure followed that trend. The 30-hour exposure to *p,p'*-DDT also reduced the expression of both NDUS1 and NDUS3. We hypothesize that DDT exposure affected first the metabolic pathways that precede the respiratory chain and later the respiratory chain itself.

Another protein with changed expression, **75 kDa glucose-regulated protein** (**GRP75**), is a molecular chaperone localized preferentially (but not exclusively) in mitochondria. This chaperone interacts with many proteins, including NDUS3 mentioned above [33]. In mitochondria, GRP75 helps to maintain mitochondrial shape and function [34, 35]. The exposure to *p,p'*-DDT strongly upregulated expression of GRP75 (see Tables 1 and 2), which may suggest that, after the exposure to DDT, mitochondria needed full support to maintain their functionality.

## Heterogeneous nuclear ribonucleoproteins

We found two members of the heterogeneous nuclear ribonucleoprotein family downregulated after exposure to *p,p'*-DDT: **heterogeneous nuclear ribonucleoprotein F** (**HNRPF**) and **heterogeneous nuclear ribonucleoprotein H** (**HNRH1**). HNRPF and HNRH1 bind to the

p53 transcript and protect it against degradation [36]; therefore, their downregulation could play a role in apoptosis initiation.

## Cytoskeletal proteins

In 2-D electrophoresis, researchers usually see the presence of fragments of cytoskeletal proteins as a sign of cell degradation [37]. Nevertheless, we found only two cytoskeletal proteins with altered expression, i.e., **vimentin** (VIME) and **keratin type II cytoskeletal** (K2C8). Both spots were at positions corresponding to their predicted size, which means that they were full-size proteins and not fragments. It is interesting to compare these data to our previous results, where we exposed NES2Y work to a non-lethal concentration of *p,p'*-DDT for 1 month and found changed expression of several cytoskeletal proteins and their fragments [6]. We hypothesize that the presence of fragments of cytoskeletal proteins does not necessarily correlate with the presence of cell death.

## Proteins involved in the maintenance of the cell morphology

We also tried to identify proteins that could play a role in the altered morphology of exposed cells. Some of the cells exposed to *p,p'*-DDT achieved a singular shape: a prolonged one, with a long thin middle section (see Fig 5). We have identified several proteins with a changed expression that could play a role in this phenomenon.

We found a downregulated expression of a fragment of ezrin after exposure to *p,p'*-DDT. **Ezrin** binds actin filaments to the plasma membrane [38], but it is difficult to evaluate the meaning of the downregulation of its fragment. **EF-hand domain-containing protein D2** (**EFHD2**), upregulated after exposure to *p,p'*-DDT, represents another protein with an altered expression that can bind actin [39]. EFHD2 participates in forming structures associated with actin, such as lamellipodia and membrane ruffles [40]. **T-complex protein 1 subunit alpha** (**TCPA**), downregulated after exposure to *p,p'*-DDT, is a part of a chaperonin called TCC (chaperonin containing t-complex polypeptide 1) responsible for the proper folding of actin and tubulin [41, 42]. **N-myc downstream-regulated gene 1** (**NDRG1**) expression can be induced by DNA damage [43], Fe deprivation [44], $Ca^{2+}$ deprivation, or hypoxia [45]. Interestingly, NDRG1 overexpression led to an altered shape of prostate cancer cells DU145 [46]. Therefore, increased expression of NDRG1 could also play a role in the altered shape of NES2Y cells exposed to *p,p'*-DDT. So could the changed protein expression of EFHD2 and TCPA.

## Proteins involved in processes connected with DNA or RNA

The exposure to *p,p'*-DDT downregulated the expression of three proteins connected with transcription, translation, or replication: PBDC1, EF2, and PCNA. **Polysaccharide biosynthesis domain containing 1 protein** (**PBDC1**, also known as CXorf26) is an unexplored protein with predicted function connected with RNA polymerase II and ribosomes [47]. **Elongation factor 2** (**EF2**) regulates peptide chain elongation on the ribosome during translation [48], but we found only a fragment of this protein downregulated. **Proliferating cell nuclear antigen** (**PCNA**) recruits participant proteins to the replication fork [49]. Downregulation of PBDC1 due to *p,p'*-DDT treatment could, therefore, negatively affect protein synthesis; downregulation of PCNA could negatively affect replication in cells. It is difficult to predict the effect of downregulation of only a fragment of EF2 on protein synthesis.

## Proteins involved in oxidative stress

The exposure to *p,p'*-DDT downregulated the expression of **biliverdin reductase A** (**BIEA**). The bilirubin/biliverdin system represents powerful protection against oxidative stress [50].

Recently, Lee and coworkers [51] described that the forced upregulation of BIEA protected INS1E rat pancreatic beta-cells from ER stress and oxidative stress. Pancreatic beta-cells have a low antioxidative defense [52]. BIEA, with better quenching ability than glutathione [51], can represent an effective defense mechanism against ROS and ER stress in pancreatic beta-cells.

The exposure to *p,p'*-DDT reduced the expression of **Ras GTPase-activating protein-binding protein 1** (**G3BP1**). G3BP1 controls an antioxidative ability of stress granules under stress conditions [53, 54]. The exposure to *p,p'*-DDT also reduced the expression of **ferritin light chain** (**FRIL**). The downregulation of FRIL can indicate that *p,p'*-DDT-treated cells are less able to quench ROS [55]. Together with the downregulation of biliverdin reductase A, these changes in protein expression suggest that after 30-hour exposure to *p,p'*-DDT, beta-cells lose their defense against oxidative stress.

## Heat shock protein 27

The exposure to *p,p'*-DDT reduced the level of **heat shock protein 27** in pancreatic beta-cells. **HSP27** inhibits apoptosis: it prevents the activation of caspase-9 by blocking the formation of apoptosome [56, 57]. The downregulation of HSP27 could be the reason why HSP27 failed to prevent the activation of caspases; activated caspase-9 was detected in cells exposed to DDT by western blot analysis.

## Conclusions

In this study, we aimed to find markers of acute toxicity of a high concentration of *p,p'*-DDT in NES2Y human pancreatic beta-cells employing 2D electrophoresis.

We have found 22 proteins that can be used as markers of acute toxicity of *p,p'*-DDT exposure in NES2Y pancreatic beta-cells. Those included proteins involved in ER stress (GRP78, and endoplasmin), mitochondrial proteins (GRP75, ECHM, IDH3A, NDUS1, and NDUS3), heterogeneous nuclear ribonucleoproteins (HNRPF, and HNRH1), cytoskeletal proteins (K2C8, and vimentin), proteins involved in the maintenance of the cell morphology (EFHD2, TCPA, NDRG1, and ezrin), proteins involved in processes connected with DNA or RNA (PBDC1, EF2, and PCNA), proteins involved in oxidative stress (BIEA, and G3BP1, and FRIL), and heat shock protein 27. The proteins we have identified may serve as indicators of p,p'-DDT toxicity in beta-cells in future studies, including long-term exposure to environmentally relevant concentrations. Also, we have found that *p,p'*-DDT-induced cell death is apoptotic cell death and that a high concentration of *p,p'*-DDT induces ER stress in NES2Y cells.

## Supporting information

**S1 File.**
(PDF)

## Author Contributions

**Conceptualization:** Nela Pavlikova.

**Methodology:** Nela Pavlikova, Jan Sramek, Michael Jelinek, Petr Halada.

**Project administration:** Nela Pavlikova.

**Writing – original draft:** Nela Pavlikova.

**Writing – review & editing:** Jan Sramek, Michael Jelinek, Petr Halada, Jan Kovar.

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
