## [Decision Letter · Decision Letter 0]

29 Jul 2020

PONE-D-20-03380

Markers of Acute Toxicity in Pancreatic Beta-Cells Exposed to Lethal Doses of the Organochlorine Pollutant DDT Determined by a Proteomic Approach

PLOS ONE

Dear Dr. Nela,

Thank you for submitting your manuscript to PLOS ONE. After careful consideration, we feel that it has merit but does not fully meet PLOS ONE’s publication criteria as it currently stands. Therefore, we invite you to submit a revised version of the manuscript that addresses the points raised during the review process.

We look forward to receiving your revised manuscript.

Kind regards,

Ch Ratnasekhar, Ph.D.

Academic Editor

PLOS ONE

Journal Requirements:

2. Please provide information about any quality control testing procedures (authentication, characterisation, and mycoplasma testing) performed on the cell line used in this work. For more information, please see http://journals.plos.org/plosone/s/submission-guidelines#loc-cell-lines.

4.PLOS ONE now requires that authors provide the original uncropped and unadjusted images underlying all blot or gel results reported in a submission’s figures or Supporting Information files. This policy and the journal’s other requirements for blot/gel reporting and figure preparation are described in detail at https://journals.plos.org/plosone/s/figures#loc-blot-and-gel-reporting-requirements and https://journals.plos.org/plosone/s/figures#loc-preparing-figures-from-image-files. When you submit your revised manuscript, please ensure that your figures adhere fully to these guidelines and provide the original underlying images for all blot or gel data reported in your submission. See the following link for instructions on providing the original image data: https://journals.plos.org/plosone/s/figures#loc-original-images-for-blots-and-gels.

5.Thank you for stating the following in the Acknowledgments Section of your manuscript:

[This work was supported by research projects UNCE 204015 and PROGRES Q36 of Charles University in Prague, Czech Republic, and by the project BIOCEV CZ.1.05/1.1.00/02.0109 from the European Regional Development Fund. The funders had no role in study design, data collection and analysis, decision to publish, or preparation of the manuscript.]

 [The funders had no role in study design, data collection and analysis, decision to publish, or preparation of the manuscript.]

Reviewers' comments:

Reviewer's Responses to Questions

**Comments to the Author**

1. Is the manuscript technically sound, and do the data support the conclusions?

Reviewer #1: No

Reviewer #2: Yes

2. Has the statistical analysis been performed appropriately and rigorously? 

Reviewer #1: No

Reviewer #2: Yes

3. Have the authors made all data underlying the findings in their manuscript fully available?

Reviewer #1: Yes

Reviewer #2: Yes

4. Is the manuscript presented in an intelligible fashion and written in standard English?

Reviewer #1: No

Reviewer #2: Yes

5. Review Comments to the Author

Reviewer #1: The manuscript “Markers of Acute Toxicity in Pancreatic Beta-Cells Exposed to Lethal Doses of the Organochlorine Pollutant DDT Determined by a Proteomic Approach” by Pavlíková et al, employs various biochemical techniques to determine the acute markers of acute toxicity in vitro.

Few are the points that must be considered to improve the overall quality of the manuscript

The term ‘lethal dose’ is incorrect to use since the dose chosen for any toxicant is always sublethal.

The authors have concluded that the detected proteins serve as the markers for acute DDT toxicity. However, they are not the exclusive markers for DDT toxicity. Maybe the authors can modify the statement or direct their conclusion towards the pathways affected following DDT exposure.

It would be advisable to add few inhibitors experiments against the targets to ascertain if the changes are solely due to DDT exposure or it’s a general response.

Are the effects observed mediated through direct binding of DDT or through its metabolites?

A thorough Grammatical proofread is required to improve the flow for better understanding

Abstract must be rewritten to highlight the rationale, potential results and findings in a precise manner

The technique however is a valuable part of the manuscript to achieve the objectives. What is the importance of acute time points and how the data obtained in the present study will be utilized for translatable value?

Please mention the culture conditions of the NES2Y cell, medium used, passage number etc.

Line 68-69 “We detected the fluorescence of cells using a FACS Calibur cytometer (Becton Dickinson, San Jose, CA, USA) channel FL2”. What does this sentence imply?

Please try to be consistent while mentioning the cell number 100 000 cells vs 1×106 cells

Please restructure the material and method section to contain the essential information like

1. What concentration of DMSO was used as solvent control?

2. Why did the authors select the time points - 24h and 30 h? Will it be feasible to assess the cellular activity at initial time points of 12 and 18 h?

Western blotting details should be explained properly in terms of methodology, antibodies and concentration.

Statistical analysis, please mention significance level.

Is there any specific mechanism the authors were interested to explore? Mentioning the requirement to kill 20% cells for 2D purpose does not make sense rationally since this percentage of cell mortality is obvious at sublethal doses of any toxicant.

Study rationale needs to be improved as in what are authors interested in and why? Is it the cell viability or the mechanism or both? This point must be clearly stated.

Please avoid repetition of sentences or same information throughout the manuscript

Reviewer #2: The manuscript by Pavlikova et al. provides an interesting perspective that highlights the acute toxic effect of DDT on pancreatic beta cells NES2Y. In the context of environmental pollutants induced diabetes, this study seems relevant.

Major concerns

1-The author should mention the doses corresponded to the comparable levels that were toxic in humans in relation to diabetes and pancreatic beta cells. If the author provides some in-vivo references or data supporting their results to show the translatability of their in-vitro findings will definitely enhance the impact of this study.

2- In figure 2, the western blots for CHOP, GRP78, cleaved caspase-9, cleaved caspase 8, cleaved caspase -7 the band thickness does not match with the densitometric quantification in figure 3. In the control groups, the bands have completely vanished.

Minor Concerns

1-The author should mention the cell viability at 30 hours.

2- In result 3.1, the author has used neutral red assay for cell viability while in figure 1, it is mentioned as propidium iodide.

6. PLOS authors have the option to publish the peer review history of their article (what does this mean?). If published, this will include your full peer review and any attached files.

Reviewer #1: No

Reviewer #2: **Yes: **Nagendra Kumar Rai

---

## [Author Response · Author response to Decision Letter 0]

25 Sep 2020

Journal Requirements:

1. 

 We have made the required changes in our manuscript.

2. 

Please provide information about any quality control testing procedures (authentication, characterisation, and mycoplasma testing) performed on the cell line used in this work. For more information, please see http://journals.plos.org/plosone/s/submission-guidelines#loc-cell-lines.

We have added this information to the section Cell culture.

3. 

We note that you have included the phrase "data not shown" in your manuscript. Unfortunately, this does not meet our data sharing requirements. PLOS does not permit references to inaccessible data. We require that authors provide all relevant data within the paper, Supporting Information files, or in an acceptable, public repository. Please add a citation to support this phrase or upload the data that corresponds with these findings to a stable repository (such as Figshare or Dryad) and provide and URLs, DOIs, or accession numbers that may be used to access these data. Or, if the data are not a core part of the research being presented in your study, we ask that you remove the phrase that refers to these data.

 We have placed the mentioned data to the Supplementary Information file and put a reference into the main text.

4.

PLOS ONE now requires that authors provide the original uncropped and unadjusted images underlying all blot or gel results reported in a submission's figures or Supporting Information files. This policy and the journal's other requirements for blot/gel reporting and figure preparation are described in detail at https://journals.plos.org/plosone/s/figures#loc-blot-and-gel-reporting-requirements and https://journals.plos.org/plosone/s/figures#loc-preparing-figures-from-image-files. When you submit your revised manuscript, please ensure that your figures adhere fully to these guidelines and provide the original underlying images for all blot or gel data reported in your submission. See the following link for instructions on providing the original image data: https://journals.plos.org/plosone/s/figures#loc-original-images-for-blots-and-gels.

We have added this information to the Supporting information file. Nevertheless, the gel pictures are partially cropped by the software we use for taking the pictures. In the case of WB membranes, we cut them into strips and expose them to different primary antibodies. Therefore, the uncropped picture is a strip and not the whole membrane.

5. 

In your cover letter, please note whether your blot/gel image data are in Supporting information or posted at a public data repository, provide the repository URL if relevant, and provide specific details as to which raw blot/gel images, if any, are not available. Email us at plosone@plos.org if you have any questions.

 We have added the required information to the cover letter.

6.

Thank you for stating the following in the Acknowledgments Section of your manuscript:

[This work was supported by research projects UNCE 204015 and PROGRES Q36 of Charles University in Prague, Czech Republic, and by the project BIOCEV CZ.1.05/1.1.00/02.0109 from the European Regional Development Fund. The funders had no role in study design, data collection and analysis, decision to publish, or preparation of the manuscript.]

 [The funders had no role in study design, data collection and analysis, decision to publish, or preparation of the manuscript.]

 We have removed the funding information from the manuscript and put it into the Cover letter for revision.

7. 

Please include captions for your Supporting Information files at the end of your manuscript, and update any in-text citations to match accordingly. Please see our Supporting Information guidelines for more information: http://journals.plos.org/plosone/s/supporting-information.

We have included captions for Supporting Information file at the end of our manuscript and updated in-text citations.

Reviewers' comments:

Reviewer's Responses to Questions

Comments to the Author

1. Is the manuscript technically sound, and do the data support the conclusions?

Reviewer #1: No

Reviewer #2: Yes

2. Has the statistical analysis been performed appropriately and rigorously? 

Reviewer #1: No

Reviewer #2: Yes

3. Have the authors made all data underlying the findings in their manuscript fully available?

Reviewer #1: Yes

Reviewer #2: Yes

4. Is the manuscript presented in an intelligible fashion and written in standard English?

Reviewer #1: No

Reviewer #2: Yes

5. Review Comments to the Author

Reviewer #1: The manuscript "Markers of Acute Toxicity in Pancreatic Beta-Cells Exposed to Lethal Doses of the Organochlorine Pollutant DDT Determined by a Proteomic Approach" by Pavlíková et al, employs various biochemical techniques to determine the acute markers of acute toxicity in vitro.

Few are the points that must be considered to improve the overall quality of the manuscript

1.1 

The term 'lethal dose' is incorrect to use since the dose chosen for any toxicant is always sublethal. 

We took, incorrectly, some deliberation in using the words "lethal dose". We have realized that according to toxicological standards, for a toxicant applied in the liquid (e.g., a cell medium), we should use the words "lethal concentration". We have established 150 μM concentration of p,p'-DDT as LC96 (it killed 96% of exposed cells) after 72 hours (see Figure S1 in Supplementary Information file). Nevertheless, because we used shorter exposure times than 72 hours in our experiments, we have replaced words “lethal dose” with words “a high concentration”.

1.2

The authors have concluded that the detected proteins serve as the markers for acute DDT toxicity. However, they are not the exclusive markers for DDT toxicity. Maybe the authors can modify the statement or direct their conclusion towards the pathways affected following DDT exposure.

When we called detected protein "markers of acute toxicity," we did not mean that the changed expression of those proteins suggested that the samples had been exposed to DDT and not to anything else. We see the changed expression of those proteins as signs of damage caused to pancreatic beta-cells by DDT. We will search for the changed expression of those proteins in samples exposed to lower, environmentally relevant concentrations DDT for a long time (months, years) to see if the environmental exposure can reach the same damage. 

To make it more clear, we have renamed our article to “Markers of acute toxicity of DDT exposure in pancreatic beta-cells determined by a proteomic approach”.

1.3

It would be advisable to add few inhibitors experiments against the targets to ascertain if the changes are solely due to DDT exposure or it's a general response. 

In this study, we did not aim to examine the exact mechanism of cell death induced by DDT. We focused on the damage caused by a high concentration of DDT and reflected by changed protein expression. 

1.4

Are the effects observed mediated through direct binding of DDT or through its metabolites?

Animals (humans included) metabolize DDT very slowly, so we do not believe that DDT metabolites can play a significant role in 24-hour or 30-hour experiments. Moreover, we think it is unlikely that pancreatic beta-cells have enzymes necessary for metabolic reactions that usually occur in the liver. In our previous studies (http://dx.doi.org/10.1016/j.envres.2015.06.046, https://doi.org/10.1038/s41598-019-54579-z), we exposed pancreatic beta-cells to lower concentrations of DDT and its metabolite DDE for 1 month. In both cases, we found only one protein that changed its expression after both DDT exposure and DDE exposure. For these reasons, we do not believe that DDT metabolites played a role in the present experiment.

1.5

A thorough Grammatical proofread is required to improve the flow for better understanding.

We have had our manuscript proofread by a native speaker.

1.6

Abstract must be rewritten to highlight the rationale, potential results and findings in a precise manner. 

We have rewritten the abstract as requested.

1.7

The technique however is a valuable part of the manuscript to achieve the objectives. What is the importance of acute time points and how the data obtained in the present study will be utilized for translatable value?

We exposed the cells for 24 hours because 24-hour exposure represents one of the standard lengths for acute toxicity tests. We exposed the cells for 30 hours because the results of 24-hour exposure failed to satisfy us (it changed the expression of only a few proteins visible at 2-DE). We chose 30 hours rather than, e.g., 36 hours as our longer exposure time for a technical reason (to have the number of survived cells high enough to provide us with enough proteins for 2-DE). 

The obtained data represent a 2-DE fingerprint of severe damage caused by high DDT concentration in beta-cells. As we described above, we will search for the changed expression of those proteins in samples exposed to lower, environmentally more relevant DDT concentrations for a long time (months, years) to see if the environmental exposure can reach the same damage.

1.8

Please mention the culture conditions of the NES2Y cell, medium used, passage number etc.

We have added this information to the new chapter in "Material and Methods" called "Cell culture." (line 81 in the manuscript with tracked changes)

1.9

Line 68-69 "We detected the fluorescence of cells using a FACS Calibur cytometer (Becton Dickinson, San Jose, CA, USA) channel FL2". What does this sentence imply? 

This sentence describes the detection of PI signal in the analyzed cells. Now the sentence is modified by the following: “For detection of PI signal (emission = 585 nm) in dead cells, we used signal detector FL2 of FACS Calibur cytometer (Becton Dickinson, San Jose, CA, USA)”. (line 99 in the manuscript with tracked changes)

1.10

Please try to be consistent while mentioning the cell number 100 000 cells vs 1×106 cells

We have unified the description of cell numbers (lines 94, 105, and 129 in the manuscript with tracked changes).

1.11

Please restructure the material and method section to contain the essential information like

1. What concentration of DMSO was used as solvent control?

2. Why did the authors select the time points - 24h and 30 h? Will it be feasible to assess the cellular activity at initial time points of 12 and 18 h?

We have added this information to the new chapter in "Material and Methods" called "Cell culture." (line 82 in the manuscript with tracked changes)

We chose 24 hours as the first exposure time. It resulted only in several changes visible at 2-DE. We prolonged the exposure to 30 hours and checked if it would lead to more protein expression changes at 2-DE and still get enough living cells to harvest proteins. We did not test shorter exposure times because, at 2-DE, it would probably fail to show any changes. 

1.12

Western blotting details should be explained properly in terms of methodology, antibodies and concentration. Statistical analysis, please mention significance level.

We have added this information to the Material and methods section (line 104, and 205 in the manuscript with tracked changes)

1.13

Is there any specific mechanism the authors were interested to explore? Mentioning the requirement to kill 20% cells for 2D purpose does not make sense rationally since this percentage of cell mortality is obvious at sublethal doses of any toxicant.

We did not intend to explore any specific cell death induction mechanism, only to check if using 2-DE can reveal any proteins that could serve as navigation for potential exploring of the mechanism of cell death induction. 

The requirement to kill 20% of cells for 2-DE purposes is a technical requirement. The chosen experiment setup provided us with enough proteins to run 2-DE. It helped us to determine a proteomic description of beta-cells exposed to a concentration of DDT so toxic that would eventually (after 72 hours) kill the exposed cells. As we describe above, we used the DDT concentration of 150 μM that killed 18% of cells after 24 hours and 96% of cells after 72 hours (S1 figure).

In our previous studies (http://dx.doi.org/10.1016/j.envres.2015.06.046, https://doi.org/10.1038/s41598-019-54579-z), we exposed beta-cells to DDT concentration of 10 μM for one month. That exposure failed to induce cell death in the exposed cells; at the end of exposure the cells showed no difference in appearance or cell growth when compared to control cells. Therefore, we do not see 20% mortality ratio as obvious at sublethal doses of any toxicant.

1.14

Study rationale needs to be improved as in what are authors interested in and why? Is it the cell viability or the mechanism or both? This point must be clearly stated.

We have rewritten the introduction and conclusions to make our study rationale clearer.

1.15

Please avoid repetition of sentences or same information throughout the manuscript

We have done our best to remove repetition throughout the manuscript.

Reviewer #2: The manuscript by Pavlikova et al. provides an interesting perspective that highlights the acute toxic effect of DDT on pancreatic beta cells NES2Y. In the context of environmental pollutants induced diabetes, this study seems relevant.

Major concerns

2.1

The author should mention the doses corresponded to the comparable levels that were toxic in humans in relation to diabetes and pancreatic beta cells. If the author provides some in-vivo references or data supporting their results to show the translatability of their in-vitro findings will definitely enhance the impact of this study.

We have added information about typical DDT levels in human serum or plasma to the introduction (lines 48 and 49 in the manuscript with tracked changes). 

The epidemiological studies usually divide people into two groups: those with diabetes and those without it, and detect the concentrations of organochlorine pollutants in their serum or plasma. Then they search for statistical differences between the concentrations of individual pollutants in people with diabetes compared to healthy people. The outcome is which pollutants can play a role in diabetes. The actual concentrations can differ among these studies. There is no clear line between what is not toxic yet and what already is.

In the present experiment, we have intentionally used the concentration much higher than the environmental levels to induce cell death. We hypothesize that long-term exposure to the environmental level of DDT could eventually lead to beta-cell death. The proteins we have found can be used as markers of the beta-cell damage in future experiments when we use lower (environmentally relevant) DDT concentrations for a long time.

2.2

In figure 2, the western blots for CHOP, GRP78, cleaved caspase-9, cleaved caspase 8, cleaved caspase -7 the band thickness does not match with the densitometric quantification in figure 3. In the control groups, the bands have completely vanished.

The densitometry represents the mean of three experimental values; the picture represents a typical example of how the western blot of the samples looked. So there can be a difference. We have added all western blots used for densitometry into the Supplement information file.

Minor Concerns

2.3 

The author should mention the cell viability at 30 hours.

We have tested the viability of the cells exposed for 30 hours using flow cytometry, and the viability was similar to that after 24 hours. But we believe that the real viability was lower than that because some of the dead cells already disintegrated into small particles (debris), and the method failed to detect them as cells. 

2.4

In result 3.1, the author has used neutral red assay for cell viability while in figure 1, it is mentioned as propidium iodide.

We used neutral red assay to test if 72-hour exposure to 150 μM DDT would kill the exposed cells. We used a neutral red assay because it is cost- and time-effective. Nevertheless, the neutral red assay cannot distinguish if the cells die more or grow more slowly due to exposure. It only determines the total number of living cells. Naturally, when only cca 4% of cells survive the experiment, this difference does not matter. In contrast, using flow cytometry with propidium iodide staining allowed us to determine the number of dead or dying cells together with the total number of cells so we could determine the concentration that killed approximately 20% of cells. 

6. PLOS authors have the option to publish the peer review history of their article (what does this mean?). If published, this will include your full peer review and any attached files.

Do you want your identity to be public for this peer review? For information about this choice, including consent withdrawal, please see our Privacy Policy.

Reviewer #1: No

Reviewer #2: Yes: Nagendra Kumar Rai

---

## [Decision Letter · Decision Letter 1]

12 Oct 2020

Markers of acute toxicity of DDT exposure in pancreatic beta-cells determined by a proteomic approach

PONE-D-20-03380R1

Dear Dr. Nela,

We’re pleased to inform you that your manuscript has been judged scientifically suitable for publication and will be formally accepted for publication once it meets all outstanding technical requirements.

Kind regards,

Ch Ratnasekhar, Ph.D.

Academic Editor

PLOS ONE

Additional Editor Comments (optional):

Reviewers' comments:

Reviewer's Responses to Questions

**Comments to the Author**

1. If the authors have adequately addressed your comments raised in a previous round of review and you feel that this manuscript is now acceptable for publication, you may indicate that here to bypass the “Comments to the Author” section, enter your conflict of interest statement in the “Confidential to Editor” section, and submit your "Accept" recommendation.

Reviewer #1: All comments have been addressed

2. Is the manuscript technically sound, and do the data support the conclusions?

Reviewer #1: Yes

3. Has the statistical analysis been performed appropriately and rigorously? 

Reviewer #1: Yes

4. Have the authors made all data underlying the findings in their manuscript fully available?

Reviewer #1: Yes

5. Is the manuscript presented in an intelligible fashion and written in standard English?

Reviewer #1: Yes

6. Review Comments to the Author

Reviewer #1: (No Response)

7. PLOS authors have the option to publish the peer review history of their article (what does this mean?). If published, this will include your full peer review and any attached files.

Reviewer #1: No

---

## [Editor Report · Acceptance letter]

16 Oct 2020

PONE-D-20-03380R1 

Markers of acute toxicity of DDT exposure in pancreatic beta-cells determined by a proteomic approach 

Dear Dr. Pavlikova:

I'm pleased to inform you that your manuscript has been deemed suitable for publication in PLOS ONE. Congratulations! Your manuscript is now with our production department. 

Kind regards, 

on behalf of

Dr. Ch Ratnasekhar 

Academic Editor

PLOS ONE